# Answering Questions on COVID-19 in Real-Time

**Jinhyuk Lee**    **Sean S. Yi**    **Minbyul Jeong**    **Mujeen Sung**
**Wonjin Yoon**    **Yonghwa Choi**    **Miyoung Ko**    **Jaewoo Kang**
Korea University
{jinhyuk_lee, seanswyi, minbyuljeong, mujeensung}@korea.ac.kr
{wjyoon, yonghwachoi, miyoungko, kangj}@korea.ac.kr

## Abstract

The recent outbreak of the novel coronavirus is wreaking havoc on the world and researchers are struggling to effectively combat it. One reason why the fight is difficult is due to the lack of information and knowledge. In this work, we outline our effort to contribute to shrinking this knowledge vacuum by creating COVIDASK,[1] a question answering (QA) system that combines biomedical text mining and QA techniques to provide answers to questions in real-time. Our system also leverages information retrieval (IR) approaches to provide entity-level answers that are complementary to QA models. Evaluation of COVIDASK is carried out by using a manually created dataset called COVID-19 Questions which is based on information from various sources, including the CDC and the WHO. We hope our system will be able to aid researchers in their search for knowledge and information not only for COVID-19, but for future pandemics as well.

## 1 Introduction

The most recent pandemic to affect humankind is the coronavirus disease 2019 (COVID-19), which has infected nearly 36 million people worldwide as of Oct 9th, 2020[2] and has led researchers and scientists to scramble to find a solution. Despite such efforts being made and experimental results being released, finding a viable treatment or vaccine for COVID-19 still seems far off.

One of the biggest hurdles to this process is the lack of knowledge regarding COVID-19 and the difficulty of finding relevant and reliable information in a timely fashion. Taking these difficulties into consideration, creating a real-time question answering (QA) system would be able to greatly aid the efforts of researchers to effectively combat the current pandemic.

Recently, QA models have made significant developments in terms of both performance and throughput. Such improvements can be attributed to the introduction of large-scale QA datasets (Hermann et al., 2015; Rajpurkar et al., 2016) and deep learning models (Seo et al., 2016; Devlin et al., 2019), while recent trends also consider the efficiency of such models (Seo et al., 2019; Dhingra et al., 2020). The development of QA models has also benefited other domains. For example, biomedical QA (Tsatsaronis et al., 2012) has seen many advances due to neural QA models (Lee et al., 2020b; Yoon et al., 2019).

Despite such progress, creating a QA model specific to COVID-19 poses several challenges. The first challenge is that there are almost no QA datasets that are tailored specifically to COVID-19 with a few recent exceptions (Möller et al., 2020). This virtually means that models will have to be evaluated in a zero-shot setting. We take the approach of using a real-time QA model (Seo et al., 2019) and evaluate its transferability from existing QA datasets to a COVID-19 dataset that we coin as the COVID-19 Questions dataset. COVID-19 Questions is created by using known facts and experimental results of COVID-19.

The second challenge is the incorporation of traditional biomedical text mining tools into existing QA models. In order to address this, we shift our focus to a key feature used in biomedical text mining: biomedical named entities. Named entities provide important information in text, and this is especially so in the biomedical domain. Taking this into account, we use BioBERT-based models (Kim et al., 2019; Sung et al., 2020) to extract and normalize biomedical named entities found in documents that contain answers. This approach would allow researchers to easily navigate named entities that

---

[1] https://covidask.korea.ac.kr
[2] https://www.who.int/emergencies/diseases/novel-coronavirus-2019

are linked to their respective Concept Unique IDs (CUIs) such as Medical Subject Headings (MeSH). To improve the stability of our approach, we incorporate a biomedical entity search engine (Lee et al., 2016) that provides relevant named entities to entity-level queries.

The contributions of our paper are three-fold. First, we present COVIDASK for real-time QA on coronaviruses to aid researchers and scientists in effectively navigating resources; second, we incorporate various techniques from traditional biomedical text mining (e.g., biomedical named entity linking (NEL)) to enhance the usability of our model; and third, we evaluate COVIDASK on our COVID-19 Questions dataset and publicly release the source code of COVIDASK and COVID-19 Questions for future work.[3]

## 2 System Desiderata

We first specify the overall goals of our system since QA is very broad and has many details to consider (e.g., types of questions, types of answers, etc.). In this section, we detail how we take such desiderata into consideration, and in Section 3 we elaborate on how we adapted COVIDASK to address each desideratum.

**Format of Questions** We are mainly interested in providing answers that are in English and in a natural language format. Natural language questions can be divided into two types: interrogative sentences (e.g., *"Where did COVID-19 originate from?"*) and short, keyword-based queries (e.g., *"COVID-19 origin"*). We cover all types of natural language questions regardless of their format, as they all reflect the need for information.

**Format of Answers** We restrict the format of answers to be contiguous n-grams within a corpus, often referred to as a "span." Although there are models that are able to generate answers that are not restricted to the given corpus (Raffel et al., 2019), the majority of models are based on "answer span extraction" (i.e., finding specific start and end indices of an answer within the given text) (Seo et al., 2016; Devlin et al., 2019). This approach makes modeling easier while maintaining effectiveness.

However, many research questions raised by researchers and scientists often require entire documents as answers rather than a simple answer span (e.g., questions regarding entire experiments or analyses regarding a certain topic). This is a more typical goal of information retrieval (IR) rather than QA. While QA models implicitly perform IR as answers are extracted from documents - especially in the case of open-domain question answering (Chen et al., 2017) - the two have been treated differently in terms of evaluation and modeling and therefore cannot be put on the same pedestal. We mainly focus on evaluating COVIDASK in a QA fashion, but also carry out IR-style evaluation via the Text Retrieval Conference COVID-19 (TREC-COVID) Challenge (Roberts et al., 2020; Voorhees et al., 2020).[4]

**Source of Knowledge** Many QA models that use unstructured text are often given a ground-truth document or paragraph that contains an answer to each question. However, it is more realistic to retrieve a relevant document first and then find an answer rather than being provided the document. This type of *Retrieve & Read* approach, popularized by Chen et al. (2017), has been termed *open-domain QA* since answers are provided directly from 5M Wikipedia documents that are not restricted to any specific domain.[5] We use the COVID-19 Open Research Dataset (CORD-19) (Wang et al., 2020) for a domain-specific corpus provided in unstructured text. Note that although COVIDASK is not an open-domain QA model, we borrow many techniques from open-domain QA since COVIDASK needs to handle a very large amount of text as well.

**Recency and Significance** An important feature that many QA models do not consider is the recency of information. Recency is important as models that use up-to-date information would be able to provide more relevant results. For instance, when asked *"Which drugs are effective for COVID-19?,"* a model that selects an answer from the latest documents would provide more value. Another interesting facet of recency is that the desired behavior of the QA model may change depending on the period of time. For example, past reports on other coronavirus-related diseases like the Middle East respiratory syndrome (MERS) do not explicitly mention COVID-19, but may nevertheless provide important clues or information that may help the understanding of COVID-19. This implies that

---

[3] https://github.com/dmis-lab/covidAsk

[4] https://ir.nist.gov/covidSubmit/

[5] The term "open" often refers to both the variety of domains and the large scales of corpora. COVIDASK's knowledge source is focused on a single domain but still uses a very large number of documents.

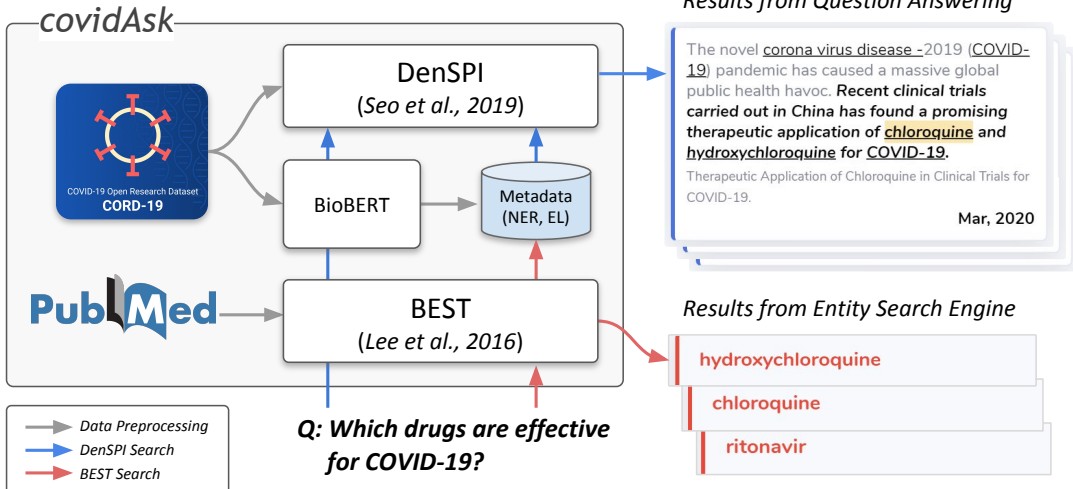

Figure 1: Overview of the COVIDASK pipeline

when using documents to find answers to questions about COVID-19, it would be appropriate to adopt a more generous stance the older a document is and vice versa.

Another important feature to take into account is the significance of the resources (i.e., how valuable or high quality the resources are). For example, answers that are chosen from research papers that have been peer-reviewed and published at reputable venues would provide more value than those from preprint servers such as medRxiv or bioRxiv. One caveat would be that this may not be applicable to literature related to COVID-19, as review processes for papers are typically lengthy and the current situation calls for the timely release of results. In order to incorporate the concepts of recency and significance into COVIDASK, we leverage the date and impact factor metadata of documents.

**User Interaction** For each extracted answer, it is essential to also provide the evidence document. This is because it is often necessary to further investigate the retrieved answers by using such evidence documents, as the provided answers themselves may not be enough. To ease such investigation, we extract biomedical named entities in our corpus and link them to their respective CUIs. This allows us to provide detailed descriptions of the named entities including their various synonyms. On top of the named entity recognition (NER) results, we incorporate an entity-level search engine called BEST (Biomedical Entity Search Tool) (Lee et al., 2016) to display important entities relevant to the question.

**Latency** Latency refers to the delay between inputting a query and receiving an answer. It is an integral aspect of QA models since such models usually deal with large and unstructured sources of knowledge, and therefore speed and efficiency are important. One way that QA models deal with the issue of latency is to retrieve only a small number of documents, thus reducing the search space and focusing only on documents that are likely to be relevant (Chen et al., 2017).

As an alternative, Seo et al. (2018, 2019) proposed to pre-index all answer candidate phrases to dense and sparse vectors and perform a maximum inner product search (MIPS) between query vectors and the phrase vectors. We adopt the approach of Seo et al. (2019) because this method allows the model to only have to run through the entire documents once regardless of the number of questions asked, making it more appropriate for real-time QA.

## 3 Architecture

The overall layout of our system is illustrated in Figure 1 and the hosted Web service is shown in Figure 2. We pre-index all phrases in research papers contained in CORD-19 (Wang et al., 2020) and use them to build the DENSPI model (Seo et al., 2019). We also highlight and use biomedical named entities in PubMed[6] for building BEST (Lee et al., 2016). For given questions, COVIDASK returns two different lists of answers from both DENSPI and BEST.

---

[6]https://www.ncbi.nlm.nih.gov/pubmed/

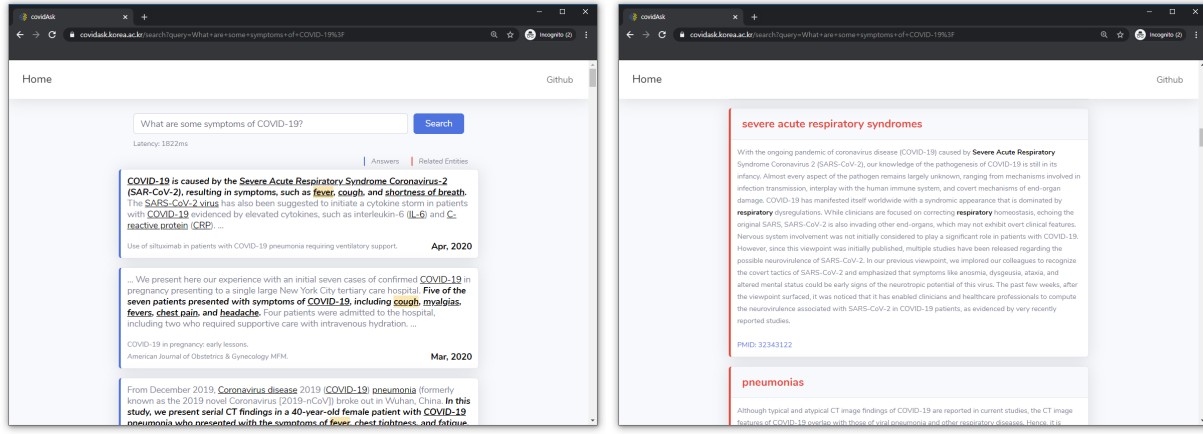

| (a) DENSPI results | (b) BEST results |

Figure 2: Overview of COVIDASK Web service. (a) and (b) each show the natural question-based and entity-based results of DENSPI and BEST for the question *"What are some symptoms of COVID-19?,"* respectively. BEST results are displayed after DENSPI results.

## 3.1 Real-Time Question Answering

We use DENSPI (Seo et al., 2019) augmented with contextualized sparse representations (SPARC) (Lee et al., 2020a). Among many open-domain QA models, DENSPI has the advantage of being able to provide answers and evidence in real-time (**Latency**).

First, each phrase vector is supervised with an extractive QA dataset such as SQuAD (Rajpurkar et al., 2016). Next, we encode all answer candidate phrases in a large document corpus into dense and sparse vectors. This essentially enables open-domain QA to be carried out in real-time as retrieving these phrase index vectors is analogous to finding answers and is significantly faster than the conventional method of reading multiple documents whenever a new question is given. Formally speaking, we directly retrieve an answer $\hat{a}$ as:

$$\hat{a} = \underset{\mathbf{x}_{i:j}^k}{\mathrm{argmax}}\, H_{\mathbf{x}}(\mathbf{x}_{i:j}^k) \cdot H_{\mathbf{q}}(\mathbf{q}) \qquad (1)$$

where $H_{\mathbf{x}}$ and $H_{\mathbf{q}}$ are encoding functions for the phrase $\mathbf{x}_{i:j}^k$ in the $k$-th document and the question $\mathbf{q}$, respectively. Due to the difficulty of directly building sparse MIPS indices, the sparse vectors are used to re-rank the top 100 phrases. SPARC (Lee et al., 2020a) is used to further enrich the sparse representations of DENSPI with lexical information, which significantly improves performance.

DENSPI provides answers in contiguous n-grams (**Format of Answers**) for natural language questions (**Format of Questions**). While the initial version of DENSPI by Seo et al. (2019) is

trained on the SQuAD dataset, some work suggests that SQuAD questions are not necessarily "genuine information-seeking questions" in that questions were made with the correct answers in mind (Lee et al., 2019). Hence, we further train DENSPI on the Natural Questions dataset (Kwiatkowski et al., 2019) which contains such information-seeking queries from search engines.

## 3.2 Biomedical Named Entity Recognition

For named entity-based user interaction, we extract and normalize biomedical entities in COVID-19 articles. First, we use BERN (Kim et al., 2019) and BIOSYN (Sung et al., 2020) for biomedical NER and NEL, respectively. Although BERN provides high-quality NER results with BioBERT (Lee et al., 2020b), its entity linking depends on many external tools and dictionaries which are insufficient for high-precision entity linking. BIOSYN, on the other hand, achieves state-of-the-art performance on many biomedical NEL datasets using a synonym marginalization algorithm. We modify BERN to use BIOSYN internally, resulting in a highly accurate NEL system. For each recognized named entity, we provide a link corresponding to its CUI whenever possible. We provide our preprocessed NER results on CORD-19 in a JavaScript Object Notation (JSON) format.[7]

We additionally use BEST (Lee et al., 2016) to provide a list of biomedical named entities that are relevant to the input questions. BEST builds an inverted index that takes biomedical named en-

---

[7]https://github.com/dmis-lab/covidAsk#data

tities in each PubMed abstract into account and returns entity-level search results. BEST is also more suitable than DENSPI for shorter keyword-based questions (e.g., *"What are the symptoms of COVID-19?"* vs. *"COVID-19 symptoms"*). Taking this into consideration, using the two models in concert would allow BEST to provide results that complement those of DENSPI.

## 3.3 Incorporating Metadata

We use of each COVID-19 article's metadata (e.g., date and venue of publication) to help users find more recent and important information (**Recency and Significance**). In our experiments, we tested re-ranking the results of DENSPI with 1) an impact factor score and 2) the time of publication of each article. However, both showed negative results in terms of our evaluation metric as balancing the importance of metadata and QA results is very difficult. We leave more effective incorporation of the metadata as our future work.

## 3.4 User Experience and Interaction

In Figure 2, we show how COVIDASK provides search results to users in its Web service. First, users can input any type of question in the search box located at the top. COVIDASK subsequently shows phrase-level answers from CORD-19 using DENSPI (blue tags) and entity-level answers from PubMed articles using BEST (red tags). For DENSPI results, each answer is highlighted in yellow and the sentence containing the answer is in boldface. When users click on a biomedical named entity (underlined), they will be redirected to the web page that contains detailed descriptions from the Comparative Toxicogenomics Database (CTD)[8] or the National Center for Biotechnology Information's (NCBI's) Taxonomy Database.[9] Other metadata (e.g., title, date, authors) are also provided whenever available.

## 4 Resources

## 4.1 Coronavirus Articles

COVIDASK mainly uses articles from CORD-19, which is a dataset that contains documents from scientific literature related to the novel coronavirus and other relevant coronaviruses and is updated on a daily basis.[10] For the experiments, COVI-

---

[8]http://ctdbase.org/
[9]https://www.ncbi.nlm.nih.gov/taxonomy
[10]https://www.semanticscholar.org/cord19/download

| Dataset | # of Examples | | |
| | Train | Dev | Test |
|---|---|---|---|
| SQuAD | 87,599 | 10,570 | - |
| Natural Questions | 79,168 | 8,757 | - |
| COVID-19 Questions | - | - | 111 |
| TREC-COVID | - | - | 30 |

Table 1: Statistics of datasets used in COVIDASK

DASK uses the April 17th, 2020 version of CORD-19 for the phrase-indexing stage of DENSPI but is updated whenever a newer version is available. We use only the abstract of each article since most key information of the articles is included in the abstract. We also conduct biomedical NER and NEL on the same version of CORD-19.

BEST, on the other hand, utilizes entire PubMed articles including articles on coronaviruses. Consequently, COVIDASK not only performs accurate QA on recent articles related to COVID-19, but also provides entity-level search results obtained from a massive amount of PubMed articles.

## 4.2 Question Answering Datasets

The main challenge of building a QA model in the COVID-19 domain is the lack of training and evaluation data. We mostly rely on existing QA datasets for training and create a separate evaluation dataset using several known facts and experimental results related to COVID-19. The statistics of the training and evaluation sets are listed in Table 1.

**Training Set** We use two extractive QA datasets for training DENSPI: SQuAD (Rajpurkar et al., 2016) and Natural Questions (Kwiatkowski et al., 2019). We use the preprocessed version of Natural Questions provided by Asai et al. (2019) where long answers are used as the paragraphs for short answers. We limit the maximum number of answer tokens to be 5 in Natural Questions following previous work on the open-domain setup of Natural Questions (Lee et al., 2019).

**COVID-19 Questions** In order to effectively evaluate COVIDASK, we manually create an evaluation dataset called the COVID-19 Questions dataset. Table 2 shows example questions from each source, and Table 3 displays the basic statistics of COVID-19 Questions.

COVID-19 Questions is composed of questions from four different sources: frequent input queries from initial users of COVIDASK, questions from

| Source | Type | Question | Answers |
|---|---|---|---|
| Query Log | Interrogative
Keyword | How long does the virus live outside
COVID-19 vaccines | [three days, 3 hours, ...]
[Chloroquine, Hydroxychloroquine, ...] |
| Kaggle | Interrogative
Keyword | What is the recommended length of quarantine?
risk factors of COVID-19 | [three weeks, 14 days, 2-week, ...]
[hypertension, diabetes, heart disease, ...] |
| CDC & WHO | Interrogative
Keyword | What diseases are caused by coronavirus?
medicines or therapies for COVID-19 | [MERS, SARS, ARDS, COVID-19, ...]
[no evidence, no vaccine, no cure, ...] |

Table 2: Sample questions from COVID-19 Questions

| | # of Samples per Type | | |
|---|---|---|---|
| **Source** | Interrogative | Keyword | Total |
| Query Log | 4 | 9 | 13 |
| Kaggle | 28 | 28 | 56 |
| CDC & WHO | 21 | 21 | 42 |
| Total | 53 | 58 | 111 |

Table 3: Statistics of COVID-19 Questions

the FAQ sections of the Center for Disease Control (CDC)[11] and the World Health Organization (WHO)[12] websites, and Kaggle's CORD-19 Challenge tasks page.[13] We categorized questions from the CDC and the WHO into one category, as the questions from the query log and Kaggle tend to be more formal and academic, whereas those from the CDC and the WHO tend to be more casual (e.g., *"What is the basic reproductive number of COVID-19?"* vs. *"What kind of hand sanitizer should I use?"*). Regardless of the grammatical details, if the sentence is in a question format we classified it as an interrogative sentence.

The key aspect that we kept in mind when creating COVID-19 Questions was *variety*. Due to the limited amount of data, it is important that various samples existed in order to effectively determine where COVIDASK performed well and where it did not. In order to test the effect of the question format, we also made sure that each interrogative sentence had a short, keyword-based counterpart and vice versa. We also varied the name used to refer to the novel coronavirus (e.g., COVID-19, SARS-CoV-2, HCoV-19).

**TREC-COVID** Although COVIDASK is designed to perform QA, it implicitly performs IR as well since the documents relevant to answers for given questions must first be retrieved. Hence, we also evaluate COVIDASK on queries from the TREC-COVID Challenge which is a competition composed of IR-based tasks. The TREC-COVID Challenge is motivated from nine research questions regarding how to use IR in a pandemic situation, and aims to find answers for questions 3 to 8 (Roberts et al., 2020). We participated in the first round which contains 30 topics (i.e., questions) in both interrogative sentence- and short, keyword-based forms.

## 5 Experiments

### 5.1 Implementation Details

For DENSPI, most hyperparameters are identical to the settings in Lee et al. (2020a) except that we stick to the dense-first search strategy in order to increase the diversity of answers. The number of CORD-19 articles (i.e., 37K abstracts) is much smaller than the number of all Wikipedia articles used in open-domain QA, and therefore we use a smaller number of centroids (1,024 centroids) for Faiss clustering (Johnson et al., 2019). For BERN and BIOSYN, we modified the original implementations to integrate the two models for biomedical NER and NEL. For BEST, we used the BEST API provided in a Python script.[14] For the submission of TREC-COVID, we used our COVID-19 Questions dataset as a validation set.

Regarding the CORD-19 articles, we mainly used the 2020-04-10 version and additionally created another version of 2020-04-10 that contains only recent articles (i.e., articles published after December, 2019) that we denote as 2020-04-10-recent. This way, we effectively reduce the search space while putting more emphasis on recent information. Additionally, we incorporate scores from Covidex (Zhang et al., 2020) in order to obtain better sparse representations for DENSPI.

---

[11]https://www.cdc.gov/coronavirus/2019-ncov/faq.html
[12]https://www.who.int/csr/disease/coronavirus_infections/faq_dec12/en/
[13]https://www.kaggle.com/allen-institute-for-ai/CORD-19-research-challenge/tasks

[14]https://github.com/SunkyuKim/BEST_API

| Model | Articles | Train | Features | Interrogative | | Keyword | | s/Q |
|---|---|---|---|---|---|---|---|---|
| | | | | $EM_{sent}@1$ | $EM_{sent}@50$ | $EM_{sent}@1$ | $EM_{sent}@50$ | |
| DENSPI + SPARC | 2020-04-10 | SQuAD | - | **0.3585** | **0.7736** | 0.0862 | 0.4483 | 1.10 |
| | 2020-04-10-recent | SQuAD | - | 0.3396 | **0.7736** | 0.1724 | 0.5172 | 0.87 |
| | 2020-04-10-recent | SQuAD | Covidex | 0.3208 | 0.7358 | 0.1897 | 0.5172 | 0.87 |
| DENSPI + SPARC | 2020-04-10 | SQuAD + NQ | - | 0.2453 | 0.6038 | 0.1552 | 0.4310 | 0.93 |
| | 2020-04-10-recent | SQuAD + NQ | - | 0.2642 | 0.6415 | 0.1552 | 0.4828 | 0.79 |
| | 2020-04-10-recent | SQuAD + NQ | Covidex | 0.2453 | 0.6038 | 0.1552 | 0.4828 | 0.79 |
| DENSPI (unpublished) | 2020-04-10-recent | NQ | - | 0.3208 | 0.6415 | 0.1897 | **0.5862** | 1.11 |
| | 2020-06-14-recent | NQ | - | 0.2453 | 0.5283 | **0.2241** | 0.5000 | 0.93 |

Table 4: Results on COVID-19 Questions

| Team | Run | Train | Features | P@5 | NDCG@10 | MAP | Bpref |
|---|---|---|---|---|---|---|---|
| sabir | sab20.1.meta.docs | - | - | 0.7800 | 0.6080 | 0.3128 | 0.4832 |
| UIowaS | UIowaS_Run3 | - | - | 0.6467 | 0.5286 | 0.2625 | 0.4686 |
| covidex | T5R1 | - | - | 0.6467 | 0.5223 | 0.1919 | 0.2838 |
| TU_Vienna | TU_Vienna_TKL_1 | - | - | 0.5133 | 0.4002 | 0.1632 | 0.2545 |
| wistud | wistud_bing | - | - | 0.4467 | 0.3362 | 0.1269 | 0.3110 |
| UB_NLP | UB_NLP_RUN_1 | - | - | 0.3800 | 0.2453 | 0.0574 | 0.2214 |
| KoreaUniversity_DMIS | dmis-rnd1-run1 | SQuAD | Covidex | 0.5867 | 0.4467 | 0.1202 | 0.2791 |
| | dmis-rnd1-run2 | SQuAD + NQ | Covidex | 0.3867 | 0.3225 | 0.0676 | 0.2339 |
| | dmis-rnd1-run3 | Manual | Covidex | 0.5867 | 0.4649 | 0.1071 | 0.2601 |

Table 5: Results on TREC-COVID Round 1. For 'Manual' submission, we manually chose better results from dmis-rnd1-run1 and dmis-rnd1-run2 for each query.

**Evaluation Metric** Many QA works use the Exact Match (EM) and F1 Score (F1) metrics between predicted answers and ground-truth answers, motivated by Rajpurkar et al. (2016). We take a more generous approach to evaluate COVIDASK since our evaluation dataset is relatively small and often has multiple answers for each question. First, we design COVIDASK to produce a sentence-level answer that contains the predicted phrase-level answer. Next, we use in-sentence EM ($EM_{sent}$) which is 1 when one of the ground-truth answers is in the predicted answer sentence and 0 otherwise. We also use top-$k$ $EM_{sent}$ ($EM_{sent}@k$) to evaluate the overall quality of the top-$k$ answers. Our evaluation metric reflects the fact that users of COVIDASK will need to read the minimal context of answers (i.e., a sentence) regardless of the correctness of the answer itself.

## 5.2 Results

Results on COVID-19 Questions are shown in Table 4. When DENSPI + SPARC is trained on SQuAD, its performance on interrogative questions is generally superior to other models. Although the performance on keyword questions when trained on both SQuAD and NQ improves with 2020-04-10 articles, it does not perform any better with 2020-

04-10-recent articles. Lastly, we include our recent ongoing effort to improve the performance of DENSPI (DENSPI (unpublished)) which is purely trained on Natural Questions. Its performance on keyword questions largely outperforms other models. As shown in the last row, we continue to update COVIDASK whenever more advanced QA models are available.

Table 5 shows the results on TREC-COVID. The results of our submissions (KoreaUniversity_DMIS) show that COVIDASK is capable of performing IR, although it is not very effective compared to other systems that are fully dedicated to IR rather than QA. This is due to the fact that QA and IR are fundamentally different tasks, as it is also possible for documents unrelated to the question itself to contain correct answers. The discrepancy between evaluation metrics used for QA and IR also imply that the two tasks put emphasis on different aspects. For more details regarding TREC-COVID Round 1 submissions, please refer to the TREC-COVID Round 1 Archive.[15]

---

[15]https://ir.nist.gov/covidSubmit/archive/archive-round1.html

| | |
|---|---|
| Q1: | Are there any medicines or therapies that can prevent or cure COVID-19? |
| A1: | **There are no specific antiviral therapies** for COVID-19. |
| A2: | Till date, **no vaccine or completely effective drug** is available to cure COVID-19. |
| A3: | **There is no cure** for COVID-19 and the vaccine development is estimated to require 12-18 months. |
| A4: | To date, **no antiviral therapy or vaccine** is available which can effectively combat the infection caused by this virus. |
| A5: | **There is no** clinically approved antiviral drug or vaccine available to be used against COVID-19. |
| Q2: | COVID-19 risk factors |
| A1: | However, this observation is not sufficient to conclude that patients **with cancer had a higher risk of COVID-19**. |
| A2: | **Abstract Cancer patients** have an increased risk of developing severe forms of COVID-19 and advanced cancer patients who are followed at home, represent a particularly frail population. |
| A3: | Conclusions: **Patients with gynecological malignant tumors** are high-risk groups prone to COVID-19 infection, and gynecological oncologists need to carry out education, prevention, control and treatment according to specific conditions. |
| A4: | First, **those with COVID-19** and preexisting cardiovascular disease (CVD) have an increased risk of severe disease and death. |
| A5: | CONCLUSION: **Patients with previous cardiovascular metabolic diseases may face a greater risk of developing into the severe condition** and the comorbidities can also greatly affect the prognosis of the COVID-19. |

Table 6: Prediction samples of COVIDASK

## 5.3 Qualitative Analysis

Table 6 displays two example questions each in interrogative sentence and keyword form and the respective sentence-level output prediction of COVIDASK that had the highest scores (from top to bottom). Within each sentence-level answer, phrase-level answers are in boldface and the linked biomedical entities are underlined. We can see that the usage of existing QA datasets indeed allows COVIDASK to effectively handle natural questions in both interrogative and keyword forms.

## 6 Discussion

Since the outbreak of COVID-19, there have been various efforts being made from multiple angles in order to handle it. Among those efforts, research focused on building effective QA & IR systems (Zhang et al., 2020; Das et al., 2020) and dataset curation for such systems (Wei et al., 2020; Gutierrez et al., 2020; Möller et al., 2020) have been especially promising as they can most directly aid researchers in narrowing the knowledge gap.

Taking into consideration the lack of datasets for COVID-19, many IR models for COVID-19 perform in an unsupervised or zero-shot setting. However, unlike IR where unsupervised models and algorithms such as the BM25 (Robertson and Zaragoza, 2009) often produce satisfying results, many QA models rely on a large amount of rich annotations (Seo et al., 2016, 2019; Devlin et al., 2019). While COVIDASK uses QA models trained

on SQuAD and Natural Questions and returns zero-shot results on COVID-19 Questions, we believe that COVIDASK could benefit from recent discoveries in unsupervised QA (Lewis et al., 2019), zero-shot QA (Brown et al., 2020), and question generation (Duan et al., 2017; Yang et al., 2017) as well. Further evaluation and fine-tuning could also be conducted using the aforementioned new COVID-19 QA datasets.

During the development of COVIDASK, we faced several challenges that are difficult to tackle considering the current state of our model. These include:

- **Answerability of questions for a large amount of unstructured text**: Although we tried to set a threshold for the scores of answers, the distribution of scores were very inconsistent across different questions (e.g., a top 1 answer with a low score is often correct).

- **Leveraging domain adapted datasets**: We tested whether using the BioASQ dataset (Tsatsaronis et al., 2012) could improve the performance of our model, but found that there were no significant improvements compared to those in Table 4.

These challenges are also actively studied in many NLP researches and we hope to tackle them in near future.

## 7 Conclusion

In this work, we presented COVIDASK in an attempt to assist the process of collecting and curating much needed knowledge by providing highly accurate answers to questions in real-time. While COVIDASK is an ongoing endeavor that will be updated accordingly, we provide a cornerstone of constructing a COVID-19 QA system and also make public the source code and evaluation dataset. We plan to further automate and refine our system so that it will be able to be adapted to provide aid in future pandemic situations as well.

## Acknowledgments

This work was supported by the National Research Foundation of Korea (NRF-2020R1A2C3010638, NRF-2016M3A9A7916996) and the Korea Health Industry Development Institute, funded by the Ministry of Health & Welfare, Republic of Korea (HR20C0021). We thank the members of Korea University and Kyle Lo (Allen Institute for Artificial Intelligence) for their insightful feedback.

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
