# OpenReview forum: "Answering Questions on COVID-19 in Real-Time"
_EMNLP/2020/Workshop/NLP-COVID — NLP-COVID19-EMNLP Oral_

### Official Review · AnonReviewer2 · 2020-09-24
**A COVID-19 QA system from two previously developed models; evaluations need to be expanded on public COVID-19 QA datasets**

**Rating:** 6
**Confidence:** 3

**Review:**

This study presents a COVID-19 QA system, COVIDASK. It uses two models previously developed (DENSPI and BEST) as the basis and adapted accordingly to the evaluation on a manually crafted COVID-19 QA dataset. The manuscript is easy to follow. I have two main questions.

First, why the indexing is at phrase-level? From Table 5 (the evaluation on the IR part), it seems that the search effectiveness on phrase-level indexing is lower than other systems indexing at passage or paragraph level.

Second, the evaluation dataset for the QA part was created by the authors themselves. Was the system evaluated on public COVID-QA datasets such as https://github.com/deepset-ai/COVID-QA and https://github.com/castorini/pygaggle/ ? The evaluation results on these datasets will be more representative since different COVID-19 QA systems can be evaluated on the same datasets.

Minor comment: please include more detail on the system such as the update frequency and any APIs supported.

---

> ### Author Response · Authors · 2020-09-27
> **Response to Reviewer2**
>
> Thank you for your comment. The reason for choosing the phrase-level QA model is that it can provide much richer information from various document sources that are seemingly unrelated to the questions. Because we do not pre-select which documents to read (and instead read all 37K abstracts in real-time), the performance on the IR benchmark that is annotated for a small subset of documents could be lower. With application of recent advancements in phrase-level open QA, we hope to close the gap while maintaining its efficiency.
>
> In the meantime, we also evaluated our model’s performance on COVID-QA (COVID-QA: A Question Answering Dataset for COVID-19, Möller et al., 2020) which has 2,019 questions with sentence-length answer annotations. Please note that our model is designed as an open QA system which assumes that the gold paragraph is not given for each question (i.e., should find an answer out of 37K passages for each question). Our model was able to achieve 8.84 F1 (top1), 20.57 F1 (top50) on this dataset. This drop is inevitable in open QA models as shown here (https://arxiv.org/abs/1704.00051, 69.5 => 27.1 EM in SQuAD), but we hope to improve the numbers soon. We will also detail the update frequency and API usage of our system in our final manuscript. Thank you.

---

### Official Review · AnonReviewer1 · 2020-09-25
**Useful tool but needs more details**

**Rating:** 6
**Confidence:** 4

**Review:**

This work presents a real-time QA tool covidAsk, which provides both the answer/supporting documents, as well as related entities. The framework mainly uses two previous works (DenSPI and BEST) with BioBERT on CORD-19 and PubMed. Despite some issues mentioned below, the paper contains substantial content that could promote the development of COVID-related QA/IR systems.


**Comments**


- The online demo was not functioning. I have tried on multiple days with multiple ips, but only the cached examples work. For example, https://covidask.korea.ac.kr/search?query=What+can+I+do+to+protect+my+family+against+COVID-19%3F will return an internal server error.

- For a *real-time* system, there should be evaluations on latency, computing resources needed, etc.

- I didn't see any evaluation on the entity search engine side. How well does it perform?

- Table 5: how should we interpret the result? From my understanding, the performance of this work is much worse compared to some of the systems. In that sense, would it make more sense to use one of the QA systems plus a regular search engine?



**Misc**
- Section 1, Paragraph 4: The development of ...

- Section 1, Paragraph 5: "The first challenge is that there are no QA datasets that are tailored specifically to COVID-19. " there is at least contemporaneous work COVID-QA (https://openreview.net/forum?id=JENSKEEzsoU).

- Section 3.4 NBCI -> NCBI

---

> ### Author Response · Authors · 2020-09-27
> **Response to Reviewer1**
>
> Thank you for the detailed comment. Due to the temporary breakdown of the BEST service, covidAsk was unable to function properly, and we have fixed the issue now. We will also detail the computational resources used for covidAsk with its efficiency (see Table 4 for second per query (s/Q) for now). The evaluation of the BEST search engine is bit challenging since the unit of retrieval is an entity not a document as in the standard ad-hoc IR. If there exists ground-truth entities for questions, it will be much easier to evaluate the quality of our search engine. For now, we refer to the original paper of BEST (https://journals.plos.org/plosone/article?id=10.1371/journal.pone.0164680) which shows manual, comparative evaluation of the entity-level search engine. Also, although our retrieval performance compared to the other search engines is lower, it does not necessarily mean that the QA performance is low. We have found that many documents that are seemingly less related to the questions often contain accurate/informative answers which shows that our model can catch long tailed information as well. We also hope to improve the quality of our phrase representation QA model with better document representation.
>
> In our final version, we will update the typos and suggestions for the reference. Thank you!

---

### Official Review · AnonReviewer3 · 2020-09-25
**Solid work with potential social impact**

**Rating:** 7
**Confidence:** 3

**Review:**

This paper describes a question-answering system that is particularly designed for COVID19-related questions. The system is capable of providing two types of answers: potential spans containing the answers to the questions and potential bio-medical entities that the query user might be interested in.

The system design of the paper is well justified, and a online demonstration of the system is provided. I just have several questions as below.

First, the "related entities" service in the online demonstration seems not working well. I tried several entities including the example shown in the paper but did not get any returned results in terms of the related entities.

Second, I like the idea of retrieving both answers and related entities for query users. Do the authors think that it is possible to use the two systems to enhance the performance of each other in the future? Now the two systems seem disjoint to me.

Finally, the authors claim that one of the highlight of the system is that it's real-time. Of course, I've tried the online demonstration and did not feel significant delay in getting answers. But it is necessary for the authors to provide evalution results on its efficiency.

---

> ### Author Response · Authors · 2020-09-27
> **Response to Reviewer3**
>
> Thank you for your comment. The ‘related entities’ service should show the returned entities from the BEST search engine, but we have found a small technical issue with BEST which was not fully functional recently (only for the last week). We have fixed the issue and it will show the related entities (mostly for short queries since BEST was not designed to deal with natural queries). As you mentioned, we are also preparing a future work that can integrate two systems to be more interactive since their results could benefit each other. Although we have included the second per query (s/Q) in Table 4, we will update the details on the efficiency of our system as well. Thank you.